# Adolescent Eating Disorder Risk in a Bilingual Region: Clinical Prevalence, Screening Challenges and Treatment Gap in South Tyrol, Italy

**DOI:** 10.3390/nu17223549

**Published:** 2025-11-13

**Authors:** Verena Barbieri, Michael Zöbl, Giuliano Piccoliori, Adolf Engl, Doris Hager von Strobele-Prainsack, Christian J. Wiedermann

**Affiliations:** 1Institute of General Practice and Public Health, Claudiana College of Health Professions, 39100 Bolzano, Italy; 2Department of Pediatrics, Hospital of Bressanone–Brixen (SABES-ASDAA), 39042 Bressanone, Italy

**Keywords:** adolescent health, SCOFF questionnaire, body image, body mass index, social media use, psychosomatic complaints, parental perception, cross-cultural screening, German, Italian

## Abstract

**Background/Objectives**: Eating disorders (EDs) in adolescents are increasingly prevalent. In South Tyrol, a bilingual region in Northern Italy, not only actual gender and age prevalences can be compared to screening rates, but even the comparability of screening tools across languages can be examined. **Methods**: A cross-sectional analysis integrated clinical registry data with representative, online school-recruited adolescents (11 to 17) self-reports. 166 clinically diagnosed cases and 1465 screened adolescents (1246 German, 219 Italian), were examined. The SCOFF questionnaire (cutoff ≥ 2 for German and ≥3 for Italian), body mass index, body image perception, psychosocial and lifestyle indicators in proxy and self-reports were examined using descriptive statistics and logistic regression. **Results**: The clinical dataset for 2024 has a prevalence rate of 0.4%. The SCOFF screening tool identified symptomatic cases in 10.6%, and an age-increasing trend among females. The overall SCOFF-prevalence did not differ between language versions, although responses to individual items varied significantly. Predictors of ED included body image, psychosomatic complaints, problematic social media use, and low social support, with differences between genders. Parents tended to underestimate their children’s perception of being “too thick.” **Conclusions**: In early adolescence, preventive strategies are needed and targeted interventions in late adolescence. For early detection and intervention, gender-sensitive prevention and active parental involvement is needed. The SCOFF questionnaire demonstrates utility across both languages, but bilingual comparison highlights the need for culturally adapted tools and cross-language validation.

## 1. Introduction

The global burden of eating disorders among adolescents has grown substantially, reflecting a growing public health challenge. Ref. [1] conducted a large-scale retrospective cohort study using the TriNetX federated research network, which aggregates electronic health records from multiple countries. Over 46 million patients aged 5 to 26 years were included in the analysis. The main finding was a dramatic increase in the prevalence of eating disorders among females aged 13 to 18 years. The authors reveal a nearly 8-fold rise between 2017 and 2022 (120/100,000 vs. 916/100,000).

Across Europe, Spain has the highest prevalence of eating disorders (1104.85/100,000), followed by Italy (1008.65/100,000), Finland (929.61/100,000), Sweden (927.22/100,000) and Austria (911.26/100,000) [2]. Further rises were reported during the COVID-19 pandemic, particularly among females [3,4]. In Italy, hospitalizations for EDs also increased during the pandemic, with significant rises in female but not male adolescents [5,6]. These developments underscore the need for early detection and prevention.

International studies highlight multiple risk factors. The German KiGGS survey found associations between ED symptoms and emotional problems, low family cohesion, and body dissatisfaction [7]. Another German study linked ED symptoms with low health literacy, negative body image, and low social status [8]. The SCOFF questionnaire [9,10] has been widely used as a screening tool, although concerns about its reliability in adolescents persist [11]. The German COPSY study reported peri-pandemic prevalence rates of 17.2% in females and 15.1% in males [12], using the SCOFF questionnaire with a cutoff < 2. An Italian study using a cutoff of three for symptomatic answers found 31.0% prevalence for EDs, with higher rates among girls, those with high BMI, and those exposed to psychosocial stressors [13].

In South Tyrol, a bilingual region in Northern Italy where about 75% of the population speaks German and 25% Italian, repeated population-based surveys have investigated youth mental health during and after the pandemic [14,15,16]. In 2025, the survey was extended to include BMI, body image, and ED symptoms using both German and Italian versions of the SCOFF. This raises the important methodological question of comparability, given the different cutoffs applied in the two versions. In this setting, questions of language-specific detection and cultural differences are of particular importance. South Tyrol’s bilingual setting is unique for detailed research on SCOFF screening instrument, since within one public health system, the instrument can be applied in both languages offering direct comparability between languages and adding additional information regarding adolescents and parental perceptions about adolescents’ body image [17,18] and lifestyle. Parental proxy- and self-reports of adolescents may differ in perceived body image, as highlighted in [19,20], where a clear underestimation of the problem by parents has been found. Detailed information on parental perceptions about their child’s body image can give implications for early detection.

Based on this background, the present study aimsto estimate the prevalence of ICD-10 diagnosed EDs among adolescents aged 11–17 years in South Tyrol;to estimate the prevalence of symptomatic EDs as assessed by SCOFF, to assess the differences between the German and Italian versions;to compare clinically diagnosed EDs and symptomatic SCOFF cases regarding age and gender;to associate psychosocial and lifestyle factors with symptomatic SCOFF scores; and to identify discrepancies between answers of parents and children in body image perception.

## 2. Materials and Methods

Clinical and survey data were collected and analyzed separately.

### 2.1. Assessment of Clinically Diagnosed EDs

This study used a cross-sectional design to provide an up-to-date snapshot of ED trends among adolescents in South Tyrol. Clinically diagnosed EDs of adolescents aged from 11 to 17 were identified through inpatient records in 2024 using ICD-10 codes F50.0 (anorexia nervosa), F50.1 (atypical anorexia), F50.2 (bulimia nervosa), F50.3 (atypical bulimia), F50.4 (binge eating associated with other psychiatric disorders), F50.5 (vomiting associated with other psychiatric disorders), F50.8 (other specified EDs), and F50.9 (unspecified ED).

Prevalence was calculated using the number of adolescents aged 11–17 years living in South Tyrol in 2024 according to official statistics from the provincial institute of statistics (ASTAT), corresponding to a total population of 39,845 individuals.

For a subsample of 25 patients with an ED diagnosis, standardized anamnesis questionnaires were available from the Department of Pediatrics, Hospital of Bressanone/Brixen. These included information on age, gender, body mass index (BMI), eating behaviors, medical risks, psychiatric comorbidities, and psychosocial problems. Due to the small sample size results are represented only in a descriptive way.

### 2.2. Screening of SCOFF Prevalence in the General Population

A population-based online survey was conducted between 17 March and 13 April 2025 using the SoSci Survey platform (Version 3.2.46, Munich, Germany). Proxy-reported data were collected for children aged 6–19 years, and self-reported data for adolescents aged 11–19 years. Recruitment was carried out through all provincial schools by contacting parents via email with a link to the questionnaire with a reminder email after two weeks. Informed consent was obtained from parents and adolescents. More than 40,000 families were invited to participate, altogether there were available 9734 proxies Out of them, 19.7% of the questionnaires were excluded preliminary due to incomplete information about demographics (age, gender, residence, parental educational status) for parent or child. Parents first completed the proxy version before adolescents could fill in the self-report form.

A total of 2554 adolescents aged 11–19 years gave their consent to participate in the survey. For the present analysis, only self-reports of adolescents aged 11 to 17 years were included, this resulted in a total of 1465 cases providing complete data regarding SCOFF questionnaire.

#### 2.2.1. Sociodemographic Measures

Collected sociodemographic variables included age, gender, parental education [21], single parenthood, and migration background.

Perceived social support was measured with the Multidimensional Scale of Perceived Social Support (MSPSS) [22]. Problematic Internet use was assessed with the German [23] and Italian [24] versions of the Generalized Problematic Internet Use Scale 2 (GPIUS-2) [25].

#### 2.2.2. Eating Disorders

Eating disorder symptoms were screened using the SCOFF questionnaire, which consists of five yes/no questions indicating symptoms of anorexia nervosa and bulimia [9,10]. In the German version [26], a cutoff of ≥2 positive answers defined symptomatic cases with a sensitivity of about 100% and a specificity of 87.5%. In the Italian version [13,27], a cutoff of ≥3 is applied with a sensitivity of 97% and a specificity of 87.3%.

For reference, the wording of the SCOFF items in English, German, and Italian versions is provided in Appendix A Table A1.

To complement SCOFF, body mass index (BMI) was calculated from self- and proxy-reported height and weight, using age- and gender-specific percentiles for underweight, normal weight, and overweight categories [28]. Body image perception was assessed with the question: “Do you think your child/you …” with five response options (is/are much too thin, is/are somewhat too thin, has/have the right weight, is/are somewhat too thick, is/are much too thick).

#### 2.2.3. Additional Psychosocial and Lifestyle Measures

Parental mental health problems were recorded via a proxy item indicating whether one or both parents had a current or past psychiatric diagnosis or psychological treatment (yes/no).

Digital media use for private purposes was assessed as the average daily time in hours adolescents reported spending on social networking, entertainment, or messaging, separate from school-related digital use ranging from 1 = “no use”, 2 = “half an hour”, 3 = “one hour”,…, until 7 = five hours or more The variable was used as a continuous variable.

Health literacy of adolescents was measured with the Health Literacy for School-Aged Children (HLSAC) 10-item scale [29], with total scores (10–40) categorized into low, moderate, and high. Average nightly sleep duration was assessed asking the usual hour of going to bed in the evening and the usual hour of getting up in the morning on school days. The number of sleeping hours was calculated. Schools burden was assessed using the question “How much do you feel stressed by school issues?” with answers on a 4-point Likert scale ranging from 1 = “not at all”, to 4 = “very much”.

The KIDSCREEN-10 questionnaire was utilized to evaluate health-related quality of life (HRQoL) [30,31]. The total score was dichotomized into low HRQoL (1) and normal/high HRQoL (0).

The BSMAS (Bergen Social Media Addiction Scale) [32] score was used to measure Problematic social media use in adolescents in self-reports with six items rated on a 5-point Likert scale ranging from 1 (very rarely) to 5 (very often). Examples of items include “How often during the last year have you used social media to forget about personal problems?”. Total possible scores range from 6 to 30, with higher scores indicating higher problematic social media use. The scale’s internal reliability was very good in the present study (Cronbach’s α = 0.88).

### 2.3. Data Analysis

Descriptive statistics of continuous variables were calculated as means (M) with standard deviations (SD), and categorical variables as absolute counts and percentages. Group differences were tested using chi-square tests for nominal variables and Mann–Whitney U tests for ordinal or non-normally distributed continuous variables. Associations were examined with Kendall’s tau-b correlations.

Reliability of multi-item scales was assessed using Cronbach’s alpha. The primary outcome variable was the dichotomized SCOFF score, indicating presence (1) or absence (0) of symptomatic EDs. For questionnaires completed in German, a cutoff of ≥2 positive answers was applied [28]; for Italian questionnaires, a cutoff of ≥3 was used [29].

To identify predictors of symptomatic SCOFF, stepwise forward multiple logistic regression was performed, reporting odds ratios (OR) with 95% confidence intervals (CI) and *p*-values. Model fit was evaluated using Hosmer–Lemeshow statistics and Nagelkerke’s R^2^. Prior to regression modelling, predictor correlations were examined, and when two predictors showed Kendall’s tau-b < 0.3, only one was retained to avoid redundancy. Receiver Operating Characteristic (ROC) analysis was used to assess sensitivity–specificity relationships of predictors.

Sample size considerations followed Bujang et al. [33], recommending ≥500 cases for observational studies in large populations.

Statistical significance was set at *p* < 0.05, *p* < 0.01, and *p* < 0.001. Analyses were conducted using IBM SPSS Statistics for Windows, Version 25.0 (IBM Corp., Armonk, NY, USA).

Since only cases with complete SCOFF-value were included in analyses, missing data were scares. There were 85 missing cases regarding the total number of psychosomatic complaints, 42 regarding self-rated BMI and 30 missing values regarding MSPSS. Some single other cases had different missing values. These cases were excluded from corresponding analyses.

## 3. Results

### 3.1. Clinically Diagnosed Eating Disorders in Adolescents

At the end of 2024, 166 adolescents aged 11–17 years in South Tyrol had a clinically diagnosed eating disorder, corresponding to a prevalence of 0.42% [0.36%; 0.48%]. The most frequent diagnosis was anorexia nervosa (F50.0, 46%), followed by unspecified ED (F50.9, 23%), atypical anorexia (F50.1, 12%), and bulimia nervosa (F50.2, 10%). All other ICD-10 diagnoses occurred in fewer than 10% of cases. As shown in Figure 1, male cases typically began at age 12 and remained relatively stable, whereas female cases started earlier and increased steadily with age.

Among a subsample of 25 patients with detailed anamnesis data (21 females, 4 males), ages ranged from 11 to 17 years, with most cases (19) occurring from age 13 onward. Twelve were underweight (BMI < 10th percentile). The most frequently reported features (Figure 2) were perfectionism, restrictive eating patterns (blacklists/low-calorie diets), and binge attacks. Underweight adolescents were more likely to report perfectionism, while those with BMI above the 10th percentile more often described binge eating and active weight control and having a food blacklist.

### 3.2. Screening Results

For the present analysis, 1465 adolescents aged 11–17 years were included (1246 completing the German version, 219 the Italian version). The sample was representative for gender and age according to provincial statistics (ASTAT). About 28.8% [26.5%; 31.2%] resided in urban areas, 11.7% [10.0%; 13.4%] lived in single-parent households, and 8.6% [7.2%; 10.2%] had a migration background.

Scale reliabilities were very good for MSPSS (α = 0.977), BSMAS (α = 0.849) and GPIUS-2 (α = 0.923), but poor for SCOFF (α = 0.569 for German, α = 0.626 for Italian).

The overall prevalence of symptomatic SCOFF was 10.6% [9.1%; 12.3%] (self-reports, ages 11–17), with clear gender differences: 5.2% [3.7%; 7.0%] in boys and 16.2% [13.6%; 19.1%] in girls. As shown in Figure 3, prevalence increased significantly with age in females but remained relatively stable in males.

When related to clinically diagnosed cases, the ratio of symptomatic SCOFF cases to ICD-diagnosed EDs showed a significant age trend in females: younger girls screened symptomatic far more often than they were clinically diagnosed, while in older girls the proportion of diagnoses to symptomatic cases increased (Figure 4). This suggests progression from subclinical symptoms to diagnosed disorders with age. In contrast, rates among boys were stable across age groups.

### 3.3. SCOFF in German- and Italian-Speaking Adolescents

Among adolescents aged 11–17 years, the prevalence of symptomatic SCOFF (cutoff: ≥2 in German, ≥3 in Italian) was 11.2% [9.5%; 13.1%] in the German version and 7.3% [4.2%; 11.6%] in the Italian version, the difference was not statistically significant, neither for girls, nor for boys.

Details on item level respnses per language are shown in Figure 5. However, item-level responses showed striking contrasts between language versions. Italian-speaking adolescents were much more likely to answer the first question positively (18.7% vs. 1.6%; *p* < 0.001), as well as the second (16.4% vs. 11.3%; *p* < 0.001) and the fourth question (9.3% vs. 15.1%; *p* = 0.009)., while German-speaking adolescents more often answered positively the fifth question (19.8% vs. 7.3%; *p* < 0.001).

These discrepancies at the single-item level suggest possible translation or cultural effects in the interpretation of SCOFF questions. Nevertheless, when using the validated cutoffs for each language version, the overall prevalence estimates were comparable across groups.

### 3.4. Body Mass Index and Body Image

Self- and proxy-reported BMI data were available for 692 males and 691 females aged 11–17 years. After excluding implausible outliers, the correlation between proxy and self-reports was high (r = 0.929, *p* < 0.001), with no significant paired difference. Therefore, only self-reported BMI is presented.

The mean BMI was 19.62 ± 3.23 for males and 19.46 ± 3.27 for females, with no significant gender difference. According to age- and gender-specific percentiles [28], 9.1% [7.1%; 11.5%] of boys and 7.8% [5.9%; 10.1%] of girls were overweight or obese, while 12.7% [10.3%; 15.4%] of boys and 15.3% [12.7%; 18.2%] of girls were underweight. BMI differed significantly between language groups for females, with Italian-speaking girls (20.76 ± 3.31) reporting higher BMI than German-speaking girls (19.22 ± 3.21; *p* < 0.001), but not for males.

Body image perception differed significantly by gender. Girls more often perceived themselves as “somewhat/much too thick,” while boys more often considered themselves “somewhat/much too thin” (*p* < 0.001 for both, proxy- and self-reports). No significant differences were observed between German- and Italian questionnaires. Correlations between proxy- and self-reports were moderate (r = 0.679, *p* < 0.001 for boys; r = 0.635, *p* < 0.001 for girls). Parents of girls underestimated their daughters’ perception of being “too thick,”. Wilcoxon tests revealed significant difference between proxy- and self-reports in girls, who perceived themselves significantly more “thicker” than their parents did (*p* < 0.001). Table 1 shows body image perception by BMI category, gender, and proxy- versus self-reports en detail. Overall, parents tended to underestimate excess weight in both boys and girls, especially in girls.

The combination of BMI, body image, and SCOFF status (Table 2) showed that adolescents in all BMI categories who perceived themselves as “somewhat/much too thick” were at elevated risk of being symptomatic on SCOFF. This pattern was particularly pronounced for overweight boys and for underweight and normal-weight girls.

Adolescents with a self-perception of being “somewhat/much too thick” were consistently more likely to score above the SCOFF cutoff, regardless of BMI status.

### 3.5. SCOFF Correlates and Predictors

Correlational analyses showed that symptomatic SCOFF was associated with a consistent set of variables across both German- and Italian-language versions, supporting comparability despite different cutoffs. The strongest associations were observed for both languages with body image “somewhat/much too thick,” number of psychosomatic complaints, problematic social media use (BSMAS) and GPIUS-2, lower perceived social support (MSPSS) for both genders, and for females with lower health-related quality of life (HRQoL), age, school’s burden and general health state. This consistency suggests that SCOFF categorization is robust across languages Full correlation coefficients stratified by gender and language and decisions about their use in the regression model are provided in Appendix B Table A2.

In multivariable logistic regression (Table 3), distinct predictors emerged by gender. Among males, a higher BSMAS score, lower perceived social support, and a body image of being “too thick” were associated with significantly greater odds of symptomatic SCOFF. Among females, predictors included a higher BSMAS score, body image “somewhat/much too thick,” a greater number of psychosomatic complaints, and a lower general health state.

Model performance was acceptable, with Nagelkerke’s R^2^ values of 0.237 (males) and 0.363 (females). The area under the ROC curve was significant for females (0.645, *p* < 0.001) but not for males, indicating stronger explanatory power of the model in females.

## 4. Discussion

This study provides the first comprehensive assessment of eating disorder symptoms among adolescents in a bilingual region of Northern Italy. Across language groups, overall SCOFF prevalence did not differ significantly when applying the validated cutoffs, although striking discrepancies at the single-item level suggest possible translation or cultural interpretation effects. Beyond prevalence, predictors of symptomatic SCOFF included body image dissatisfaction, psychosomatic complaints, reduced perceived health, lower social support, and higher social media use, with gender-specific patterns indicating different risk constellations for boys and girls. Finally, parents, particularly of girls, tended to underestimate their children’s perception of being “too thick,” underscoring the importance of integrating both adolescent and parental perspectives when addressing early signs of eating disorders.

### 4.1. Clinical vs. Screening Prevalence in a Bilingual Context

The comparison of clinical diagnoses with population-based screening reveals a substantial treatment gap in adolescent eating disorders. In late 2024, clinically diagnosed cases in South Tyrol were 0.4%, while SCOFF screening identified 10.6% of adolescents as symptomatic. This aligns with evidence that only a minority of symptomatic adolescents are detected in clinical settings [11,12], and is consistent with international findings reporting clinical prevalence of 0.4–5% versus screening-based rates between 10% and over 50% [34,35]. Age and gender patterns showed distinct trends: in boys, diagnosed and symptomatic cases remained stable across adolescence, whereas in girls, symptomatic SCOFF increased with age and the proportion of clinical diagnoses compared to symptomatic cases rose in later adolescence. This indicates that in females, symptoms often emerge early but may take years to reach clinical threshold, highlighting the need for early prevention in younger groups and timely treatment access for older adolescents. These findings align with international reports of rising female adolescent ED incidence during the pandemic [3,4,5,6,17] and confirm that South Tyrol follows the trajectory observed in other high-income countries. Many of these cases likely fall into the category of subthreshold or partial eating disorders, which are clinically relevant and predictive of progression [36,37]. Given the well-documented long-term risks including psychiatric comorbidity, high-risk behaviors, and adverse weight outcomes [38,39], reducing the gap between symptomatic and diagnosed cases is an urgent public health priority.

Findings underscore the importance of early detection and prevention of eating disorders. A recent review [40] supports the shift toward earlier intervention, particularly when programs incorporate media literacy, self-esteem building, and peer influence components. The literature provides broad evidence of eating disorder prevention in different settings and programs [41,42,43,44,45]. Raising awareness of eating disorders among pediatricians and general practitioners, integrating mandatory prevention workshops and removing potential harmful activities from school curricula could be future strategies.

The bilingual administration of SCOFF provides important insights into the cross-cultural applicability of screening tools. After applying validated cutoffs (≥2 in German [26], ≥3 in Italian [27]), prevalence estimates did not differ significantly between language groups, suggesting that both versions can be used for population surveillance. This finding is consistent with validation studies in French, Spanish, Persian, Chinese, and Japanese populations, where overall prevalence rates were also comparable across language groups when culturally adapted cutoffs were applied [46,47,48,49,50]. However, striking discrepancies emerged at the item level: Italian-speaking adolescents in South Tyrol more frequently answered the first SCOFF question positively than German-speaking adolescents. This discrepancy may be due to the different wording. While in the German version they were asked whether they endorsed vomiting, in the Italian version they only were asked whether they felt uncomfortably full. Italian speaking adolescents were more likely to lose control over eating, whereas German-speaking peers more often endorsed food dominating life. Such variation has been reported elsewhere, for example, in Japanese and Danish studies, where differences in endorsement of vomiting and weight loss items were linked to translation nuances and cultural attitudes toward eating behaviors [50,51]. These discrepancies may therefore reflect both linguistic differences and genuine cultural variation in symptom expression.

Similar concerns exist about SCOFF’s low internal consistency in adolescent populations, with reliability coefficients often below acceptable thresholds [50,52]. A systematic review confirmed that while SCOFF performs adequately for population-level surveillance, its diagnostic accuracy is limited and should be interpreted with caution [53,54]. The low reliability of SCOFF (α = 0.569 for German, α = 0.626 for Italian) has even been found in [11]. Taken together, these results highlight the importance of linguistic validation, cultural adaptation, and cautious interpretation when using the SCOFF in diverse populations. A cross-language validation of the instrument can provide further insights for the application of the instrument in future. Further research should explore differential item functioning across languages and examine whether harmonized cutoffs or complementary screening items could enhance comparability and reliability. Further, the exploration of additional factors pointing to symptoms of eating disorders should be examined, best in controlled trials comparing the answers of general populations to the answers of clinically diagnosed cases.

### 4.2. Psychosocial and Family-Related Determinants

The analysis identified psychosocial and lifestyle factors linked to symptomatic SCOFF scores. Across language groups, body image dissatisfaction, psychosomatic complaints, problematic social media use, and low perceived social support were significant correlates, with gender-specific patterns emerging. Among boys, symptomatic SCOFF scores were strongly linked to excessive social media use, low social support, and body dissatisfaction. Among girls, symptomatic cases were associated with psychosomatic complaints and poorer perceived health. These results are in line with other Italian surveys [35]. They also reflect broader international findings that body image dissatisfaction is a robust and rising risk factor for disordered eating in adolescents [19,20,55], and that psychosomatic complaints and social media exposure amplify this risk [19,56]. Taken together, these findings underscore the importance of digital media influences, psychosomatic strain, and body dissatisfaction as predictors of symptomatic eating behaviors, and suggest that interventions focusing on media literacy, health literacy, stress regulation, and peer and family support may help reduce risk.

The comparison of self-reports and proxy-reports revealed a discrepancy in body image perception. Adolescents, particularly girls, were more likely to consider themselves “too thick” compared with their parents’ perceptions. This mismatch suggests a potential under-recognition of risk by caregivers, which may delay support or help-seeking. Such parental underestimation of body image concerns has been documented elsewhere and is especially pronounced in female adolescents [19,20].

Parental involvement therefore remains essential in the prevention and treatment of eating disorders [36,37,38]. The present findings show that parental perception alone is insufficient to detect early risk. Integrating both adolescent self-perceptions and caregiver perspectives into early screening and counselling can provide a more accurate basis for targeted early intervention.

### 4.3. Implications for Local Policies

The findings have significant implications for health and education policy in South Tyrol. The notable discrepancy between symptomatic and clinically diagnosed eating disorders underscores the necessity for a coordinated prevention framework that integrates healthcare, education, and social services [57]. Early identification of risk indicators, such as body dissatisfaction, psychosomatic stress, and problematic social media use, can be enhanced through structured screening and referral pathways within schools, community pediatric services, and youth counseling centers. Given that symptoms may manifest well before adolescence, preventive measures should commence early in life, beginning with preschool and primary school programs that promote positive body image, emotional regulation, and healthy relationships with food.

The cultural complexity of South Tyrol, including multicultural environment, presents both challenges and opportunities. The region’s linguistic and cultural diversity may influence adolescents’ understanding and reporting of eating-related symptoms. Variations observed at the single-item level suggest that cultural norms and language nuances shape both symptom expression and help-seeking behavior. Tailored prevention messages, accessible in both languages and sensitive to the growing multicultural context, could enhance awareness and reduce stigma across communities.

Integrated, cross-sectoral approaches are crucial to safeguarding children and adolescents at risk. Schools can play a pivotal role through teacher training, digital media education, and collaboration with school psychologists, dietitians, and primary care providers [58]. Health authorities could support ongoing surveillance of adolescent mental health and eating behaviors as part of a broader child health monitoring system. Collaborative efforts among the Departments of Health, Education, and Social Policy would facilitate the creation of a sustainable regional strategy aimed at early detection, family engagement, and timely access to multidisciplinary care.

### 4.4. Methodological Considerations and Future Research

The findings must be interpreted considering both methodological strengths and limitations. A major strength of the study is the integration of clinical registry data with population-based screening in a bilingual region, providing a unique opportunity to compare diagnosed cases with subclinical symptoms. The large and representative sample increases generalizability within South Tyrol.

Limitations include the cross-sectional design, which precludes conclusions about temporal or causal relationships, and the reliance on self-reported measures, which are susceptible to recall and reporting biases. A further limitation is about screening, the SCOFF questionnaire demonstrated poor internal consistency in both the German and Italian versions, in line with previous reports questioning its reliability among adolescents [18,32,33,34]. Concerns about false-positive results are well founded, as such misclassification can cause unnecessary anxiety, stigma, and medical referrals [32]. The smaller Italian-speaking sample limits statistical power.

Future research in South Tyrol should focus on the validation of bilingual screening tools by directly comparing symptomatic SCOFF scores in clinical and healthy populations. Longitudinal studies are needed to examine the progression from subclinical symptoms to clinically manifest eating disorders, particularly in girls. An extension of the analyses to young adults could provide further important insights. Mixed methods approaches, including qualitative work, could provide insights into cultural differences in how eating disorder symptoms are expressed and understood. Finally, the integration of digital health data may offer opportunities to explore the influence of online behaviors on body image and eating patterns, complementing traditional survey-based methods. Combining SCOFF with additional targeted items, such as those assessing perfectionism, restrictive eating, binge attacks, and body image distortion, may enhance its screening value in adolescents.

## 5. Conclusions

This study identifies a large treatment gap in adolescent eating disorders in South Tyrol: while only 0.4% of adolescents received a clinical diagnosis, 10.6% screened positive using the SCOFF questionnaire. The prevalence of symptoms and diagnoses in girls increased with age, but the ratio of diagnoses to symptoms was higher among girls in later adolescence, underscoring the necessity for early prevention in younger cohorts and timely intervention for older adolescents. The German and Italian versions of the SCOFF yielded comparable prevalence estimates with validated cutoffs; however, item-level differences underscore the importance of linguistic validation and cultural adaptation. Findings advocate for early, family-inclusive, gender-sensitive prevention and intervention, enhanced cross-cultural screening tools.

## Figures and Tables

**Figure 1 nutrients-17-03549-f001:**
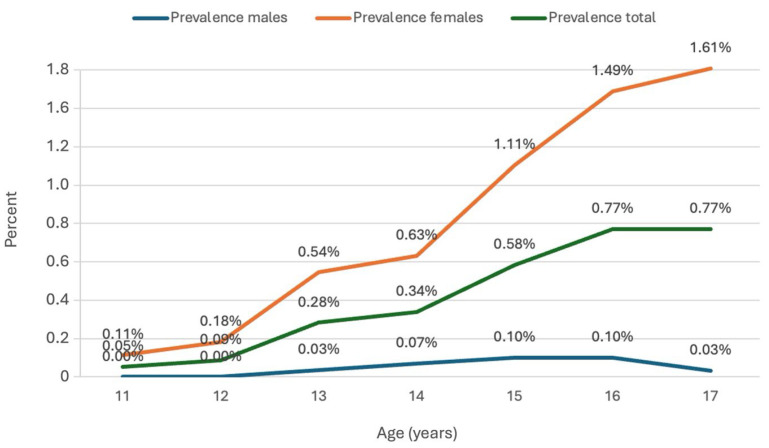
Line graph showing the age-specific prevalence of clinically diagnosed eating disorders (ICD-10 codes F50.0–F50.9) among male and female adolescents aged 11–17 years in South Tyrol (2024).

**Figure 2 nutrients-17-03549-f002:**
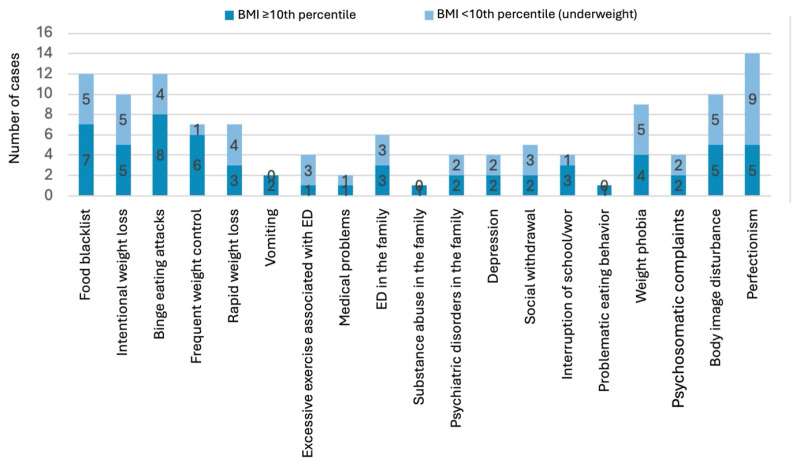
Clinical features from anamnesis questionnaires (subsample of 25 adolescents) in adolescents with diagnosed eating disorders, stratified by BMI percentile.

**Figure 3 nutrients-17-03549-f003:**
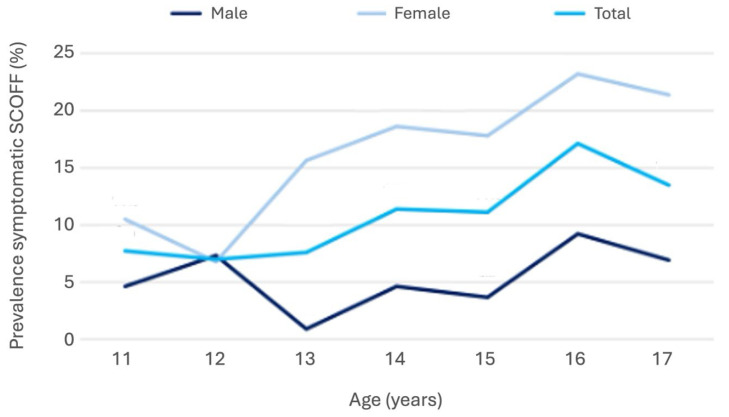
Line graph showing the prevalence of symptomatic SCOFF cases (cutoff ≥ 2 for German, ≥3 for Italian) among male and female adolescents aged 11–17 years in South Tyrol.

**Figure 4 nutrients-17-03549-f004:**
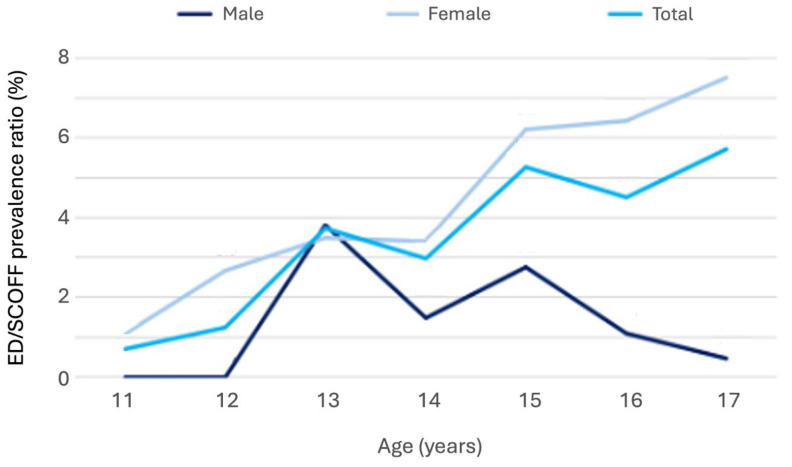
Line graph showing the proportion of clinically diagnosed ED cases relative to the prevalence of symptomatic SCOFF cases among male and female adolescents aged 11–17 years in South Tyrol.

**Figure 5 nutrients-17-03549-f005:**
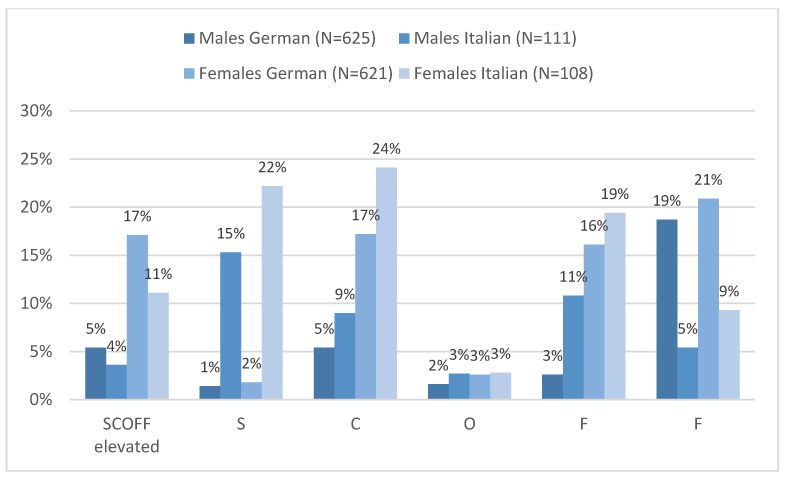
Bar charts show the prevalence (%) of elevated SCOFF scores cutoff (≥2 in German, ≥3 in Italian) and positive responses to the five SCOFF items among adolescents aged 11–17 years in South Tyrol, stratified by gender and language version (German vs. Italian).

**Table 1 nutrients-17-03549-t001:** Body image by BMI category, gender, and proxy- versus self-report.

**Males**	**Underweight (%)**		**Normal (%)**		**Overweight (%)**	
	**Proxy 11–17**	**Self 11–17**	**Proxy 11–17**	**Self 11–17**	**Proxy 11–17**	**Self 11–17**
Much too thin	3.1	4.1	1.1	1.6	0.0	0.0
Somewhat too thin	39.2	37.1	14.3	16.6	0.0	0.0
Right weight	56.7	55.7	76.7	72.2	40.0	27.7
Somewhat too thick	0.0	2.1	7.4	8.5	50.8	61.5
Much too thick	1.0	1.0	0.5	1.0	9.2	10.8
**Females**	**Underweight (%)**		**Normal (%)**		**Overweight (%)**	
	**Proxy 11–17**	**Self 11–17**	**Proxy 11–17**	**Self 11–17**	**Proxy 11–17**	**Self 11–17**
Much too thin	4.0	3.2	0.0	0.0	0.0	1.7
Somewhat too thin	19.2	23.2	3.5	3.9	0.0	0.0
Right weight	69.6	62.4	80.1	71.1	33.9	16.9
Somewhat too thick	6.4	9.6	15.7	22.9	57.6	76.3
Much too thick	0.8	1.6	0.7	2.2	8.5	5.1

**Table 2 nutrients-17-03549-t002:** SCOFF results by BMI and body image perception (self-reports, ages 11–17).

	Male		Female	
	SCOFF Normal	SCOFF Symptomatic	SCOFF Normal	SCOFF Symptomatic
Total	94%	6%	83%	17%
	Underweight			
Much too thin	4 (100%)	0	3 (100%)	0
Too thin	31 (100%)	0	22 (100%)	0
Normal	50 (98.0%)	1 (2%)	62 (91.2%)	6 (8.8%)
Somewhat too thick	1 (100%)	0	5 (45.5%)	6 (54.5%)
Much too thick	0	1 (100%)	1 (50%)	1 (50%)
	Normal weight			
Much too thin	8 (100%)	0	0	0
Too thin	82 (96.5%)	3 (3.5%)	19 (90.5%)	2 (9.5%)
Normal	380 (97.2%)	11 (2.8%)	345 (91.8%)	31 (8.2%)
Somewhat too thick	40 (81.6%)	9 (18.4%)	76 (62.3%)	46 (37.7%)
Much too thick	4 (80%)	1 (20%)	3 (27.3%)	8 (72.7%)
	Overweight			
Much too thin	0	0	0	1 (100%)
Normal	17 (94.4%)	1 (5.6%)	8 (100%)	0
Somewhat too thick	34 (87.2%)	5 (12.8%)	28 (65.1%)	15 (34.9%)
Much too thick	2 (33.3%)	4 (66.7%)	1 (50%)	1 (50%)

**Table 3 nutrients-17-03549-t003:** Logistic regression for predictors of symptomatic SCOFF among adolescents aged 11–17 years (self-reports).

	Males		Females	
	OR [95% CI]	*p*-Value	OR [95% CI]	*p*-Value
BSMAS score	1.11 [1.03; 1.20]	0.009	1.12 [1.06; 1.18]	<0.001
MSPSS score	0.75 [0.60; 0.92]	0.006		n.s.
Body image too thick	11.61 [5.57; 24.19]	<0.001	5.34 [3.25; 8.76]	<0.001
Number of psychosomatic complaints			1.27 [1.12; 1.44]	<0.001
Age				n.s.
General health state			1.64 [1.20; 2.24]	0.002
Nagelkerkes’ R^2^	0.237		0.363	
Hosmer-Lemeshow		n.s.		n.s.
Area under the ROC curve	0.511	n.s.	0.645	<0.001
N	652		620	

## Data Availability

The datasets generated and analysed during the current study are available from the corresponding author upon reasonable request. The data are not publicly available due to privacy restrictions.

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
