# Peer review of "Adolescent Eating Disorder Risk in a Bilingual Region: Clinical Prevalence, Screening Challenges and Treatment Gap in South Tyrol, Italy"

_nutrients, 2025, doi:10.3390/nu17223549_

Round 1
Reviewer 1 Report
Comments and Suggestions for Authors
Dear authors,
This is a well-designed epidemiological study addressing a relevant topic: eating disorder (ED) screening and cultural comparability of the SCOFF questionnaire in a bilingual adolescent population. The sample size, data integration, and bilingual focus make this paper potentially publishable.
However, the manuscript still needs refinement in clarity, precision, and organization. Certain methodological details are missing, and the discussion contains repetitive or overly general paragraphs. Below are specific, line-by-line comments.
Comment 1 – Abstract (L13–35)
The abstract is informative but overly dense. It should be rewritten for brevity and clarity:
-
Simplify the first two sentences and clearly state background, objectives, and design.
-
Clarify what “representative population-based online survey” means (mention school recruitment).
-
Remove numerical details by gender and age—keep only main findings.
-
Replace interpretive language (“necessitates further validation”) with descriptive phrasing.
-
End with a concise sentence linking main result and implication (e.g., “bilingual comparison highlights the need for culturally adapted tools”).
Comment 2 – Introduction (L41–79)
The introduction provides solid background but could be more focused and contextualized:
-
Start with a clearer magnitude of the problem (global prevalence 1–5%) and cite recent reviews (2023–2025).
-
Briefly explain why South Tyrol’s bilingual setting is unique for public health and ED research.
-
Add a transitional sentence linking parental perception to study goals.
-
Rewrite the list of research questions in compact bullet style (“to estimate…”, “to compare…”) to follow Nutrients formatting standards.
Comment 3 – Materials and Methods (L81–188)
The methodology is well-structured but still lacks clarity on several operational details:
-
Specify whether clinical and survey data were analyzed jointly or separately.
-
Explicitly mention the small subsample (n=25) as a limitation for inferential analysis.
-
Add response rate and inclusion/exclusion criteria.
-
Clarify how the “burden from external stressors” index was validated or referenced.
-
State the psychometric source and permission for the SCOFF Italian version.
-
Explain how “digital media use” was used in regression (continuous vs. categorical).
-
Summarize handling of missing data and remove redundant text about sample size formulas.
Comment 4 – Results (L189–322)
Results are comprehensive but overly detailed and sometimes interpretive:
-
Condense prevalence description (L190–193) into one clear statement.
-
Include CIs or SEs for representativeness.
-
Move the note on SCOFF reliability (α=0.57–0.63) to Discussion.
-
Reorganize comparison between German and Italian SCOFF items, emphasizing clinical interpretation over descriptive repetition.
-
Simplify body image results—highlight only significant gender differences.
-
Move interpretive lines (L298–301) to Discussion.
-
Complete Table 3 with all odds ratios and p-values.
Comment 5 – Discussion (L323–445)
The Discussion is rich but repetitive and needs synthesis:
-
Shorten the first paragraph; it repeats Results.
-
Condense prevention literature (L360–372) to key meta-analyses.
-
Strengthen the psychometric discussion with direct mention of SCOFF reliability (α=0.57/0.63) and propose cross-language validation.
-
Avoid repetition in psychosocial factors; reduce literature review tone.
-
Clarify that the cited attachment study (L428–433) is on a younger age group.
-
Conclude this section with a clear, practical takeaway (e.g., include parents in early screening).
Comment 6 – Methodological Considerations (L446–477)
This section appropriately highlights limitations but could be more concise:
-
Rephrase “Even the number of Italian participants was small…” → “The smaller Italian-speaking sample limits statistical power.”
-
Combine recommendations on improving SCOFF into one short paragraph.
-
Merge this section with Conclusions to avoid redundancy.
Comment 7 – Conclusions (L478–496)
The conclusions are well supported but too long and partially repetitive.
Focus on three concise take-home messages:
-
Treatment gap: Only 0.4% clinically diagnosed vs. 10.6% symptomatic.
-
Validation need: Comparable prevalence across languages, but item-level differences justify further psychometric validation.
-
Prevention: Family-inclusive, gender-sensitive, culturally adapted approaches are essential.Best Regards
Author Response
Please find comments attached

Reviewer 2 Report
Comments and Suggestions for Authors
How can this data support local actions/strategies/policies for the protection of children and adolescents with eating disorders?
There is an issue that seems interesting to me: attention to the child, from birth to the preschool phase. There is little discussion about this phase of life, but, even if it is not the focus of this article, it can, in some way, guide future studies and related debate.
Given the cultural complexity of the region, are there any differentiating factors perceived in this data?
What is the possibility of combined and ongoing actions, involving various sectors (e.g., health, education, school, social assistance, etc.) for the protection of these children and adolescents?
To what extent can local schools, more specifically, contribute with strategies?
Author Response
Please find comments attached

Round 2
Reviewer 1 Report
Comments and Suggestions for Authors
Dear authors,
After carefully reviewing your revised version, I can confirm that you have addressed the reviewers’ comments in a consistent and thorough manner. The manuscript now reads with greater clarity and coherence, and the scientific content has been considerably strengthened. The abstract has been simplified and communicates the study objectives and implications with precision, the introduction situates the problem effectively with updated epidemiological data, and the methodological section now provides sufficient transparency regarding data sources, inclusion criteria, and statistical procedures. The results are clearly organized, supported by well-presented tables and figures, and the discussion is more balanced, avoiding redundancy while offering a concise interpretation consistent with the findings. The conclusions emphasize the key messages in a focused and realistic way. Overall, the manuscript now meets the quality standards expected in Nutrients and conveys a strong contribution
Kind regards,
Reviewer 2 Report
Comments and Suggestions for Authors
The modifications made in the new version are sufficient